# Age Three: Milestone in the Development of Cognitive Flexibility

**DOI:** 10.3390/bs14070578

**Published:** 2024-07-08

**Authors:** Chufan Wan, Hui Cai, Fuhong Li

**Affiliations:** School of Psychology, Jiangxi Normal University, Nanchang 330022, China; wanchufan223@foxmail.com (C.W.); caihui.123@foxmail.com (H.C.)

**Keywords:** cognitive flexibility, rule induction, switch, inhibition, generalization

## Abstract

Although the cognitive flexibility (CF) of preschool children has been extensively studied, the development of CF in children around three years old is unclear. This study aimed to investigate the CF of three-year-olds in a stepwise rule-induction task (sRIT) comprising nine steps in which children are encouraged to switch attention to a new rule and then implicitly inhibit the old one. A pair of boxes was displayed at each step, and children aged 2.5 to 3.5 years were asked to select the target. When children learned a rule (e.g., the shape rule), they were encouraged to switch rules through negative feedback. The results showed that most children (81.10%) passed at least one of the two sets of the sRIT, and children over the age of three years performed better than those under three years. Additionally, a positive correlation existed between rule switching and rule generalization, whereby the old rule was implicitly inhibited. These findings indicate that age three might be a milestone in the development of CF, and inhibitory control might play a vital role in rule switching.

## 1. Introduction

Executive function, also known as executive or cognitive control, is the conscious control over thoughts and actions [1,2]. Cognitive flexibility (CF) is a critical component of executive function, which refers to the ability to actively adjust thoughts and behaviors in response to external changes or feedback [3,4]. CF enables one to think divergently, change one’s perspective, and adapt to a continuously changing environment [5,6]. Evidence shows that CF occurs in early childhood and develops rapidly between the ages of three and six [7,8].

The Dimensional Change Card Sort (DCCS) task is a widely used experimental task for measuring CF in children [9,10]. In the standard version of the DCCS task, children are first asked to sort a series of bivalent test cards (e.g., red boats and blue rabbits) using one rule (e.g., color) in the pre-switch phase and then the second rule (e.g., shape) in the post-switch phase [11]. Previous studies have found that most four- to five-year-old children can switch flexibly, whereas three-year-olds tend to make the perseverative error (PE) of continuing to sort cards according to the old rules [11,12,13].

Various theories have been proposed to account for the remarkable PE in three-year-olds, including cognitive complexity and control (CCC), attentional inertia, negative priming, competitive memory, goal neglect, and representational redescription theories [14,15,16,17,18,19]. Among these, the attentional inertia theory posits that PE is caused by difficulty in inhibiting the tendency to think of stimuli in a particular way and continuing to attend to the pre-switch dimension. The negative priming account emphasizes the role of inhibitory control, and argues that the irrelevant dimension inhibited in the pre-switch phase is not easily reactivated in the post-switch phase. These two theories are supported by the results of a meta-analysis that indicated that PEs in three-year-olds were best interpreted from the perspective of inhibitory control [20].

In addition to the DCCS task, other tasks have been used to explore the CF of children aged three to five [11,21,22]. The tasks in these studies typically require children to follow lengthy instructions, demonstrate good selective attention (e.g., focus on the shape of a stimulus while ignoring its color), and resolve stimulus–stimulus conflicts [23]. For example, in the Shape School task, a colorful storybook portrayed a schoolyard with colorful circles and squares [24,25]. Three conditions were used: control, inhibition, and switching. In the “control” condition, children were told that the names of the shape stimuli attending school were their colors and were asked to name the colors (blue, yellow, red) of the stimuli. In the “inhibit” condition, some stimulus figures were shown with happy faces, whereas the remaining had a sad or frustrated expression. Children were told to name the colors of the happy-faced stimuli, while inhibiting those associated with the unhappy-faced stimuli. In the “switch” condition, some stimulus figures were wearing hats. The child was told that now in this classroom, the names of the figures wearing hats were their shape (e.g., square or circle) and that the names of the figures without hats retained their color (e.g., blue, yellow, or red). The Shape School task is deemed unsuitable for children aged three years because of the comparatively high requirements of inhibition in the “switch” condition.

The A-not-B task, a classical paradigm used to investigate the development of many critical early cognitive abilities, and involves hiding a desirable object at location A for several trials and then hiding it at a new location—B. Several studies have used this task to demonstrate the emergence of CF in infants during their first year of life [26,27]. However, it is unknown how CF develops around the age of three [28,29]. Blakey et al. [30] adopted the Switching, Inhibition, and Flexibility Task (SwIFT), which consists of two conditions—conflict and distraction—to investigate CF in 2.5- to 3-year-old children. The conflict condition was identical to the standard version of the DCCS task, whereas in the distraction condition, the post-switch stimuli did not have a response conflict, because the incorrect response option did not match the prompt image on the previously relevant dimension. Therefore, the children did not need to inhibit the previous rule when implementing the new rule. They found that children’s CF in switching rules under the distraction condition developed around the age of three years [30]. A recent study also revealed that even two-year-old children could successfully switch their attention in a nonverbal visual tracking task when cues consistent with the target stimulus were provided before rule switching, thereby fulfilling the requirement of reducing the inhibition of the old rule [31].

Taken together, existing studies have mainly investigated the development of CF in 3- to 4-year-old children. However, the majority of experimental tasks, such as the DCCS task, Shape School, and Conflict SwIFT, are challenging for children under the age of three who have yet to develop inhibition. The failure of young children in these tasks may be because they are required to inhibit not only the interference of the pre-switching rule but also the dimensional conflict caused by the non-target stimulus. In contrast, 2- to 3-year-old children can switch from the old rule to the new rule for tasks without demanding rule inhibition [30,31].

The issue to be addressed in our study is how CF develops in 3 year olds if rule inhibition is not excluded from the task, but merely separated from rule switching in the temporal dimension. To achieve this, a stepwise rule-induction task (sRIT) was devised and modified from the DCCS task. In the sRIT, the post-switching phase was divided into two sub-processes: rule switching and rule inhibition. After learning a rule, children were first instructed to switch to the new rule while the pre-switching rule (i.e., the old rule) remained, and then the old rule was inhibited implicitly. The sRIT comprises three phases with nine steps. In the first phase, children select a target based on their preferred rules. For instance, they may prefer a shape rule and assume that boxes with cylindrical shapes contain targets. In the second phase, they shift their attention to the new rule (e.g., pattern/color) through negative feedback, while the previous rule (e.g., shape) remains intact; therefore, inhibiting the old rule is not necessary for rule switching. The third phase involves inhibiting or ignoring the old rule and generalizing the new rule to other stimuli of different shapes.

Two predictions were made. First, the sRIT is a rule-learning task similar to the Wisconsin Card Sorting Test (WCST), including rule formation, switching, and generalization phases. Children switch rules in the fifth step of phase 2, but do not need to inhibit the old rules of phase 1. In other words, the new rule does not conflict with the old rule, thereby reducing the cognitive demands of inhibition in younger children. Therefore, we predicted that most children aged three or younger would succeed in switching rules. Second, we expected that, like the DCCS, CF in the sRIT would improve with age [12,19]. To discover subtle developmental changes between the ages of 2.5 and 3.5 years, we used minor age bands (i.e., three months) [32].

## 2. Materials and Methods

### 2.1. Participants

Sixty children were recruited randomly from two preschool and nursery classes (M = 38.26 months, 33–42 months, 32 girls), and each child performed two sets of the sRIT, with a 5–10-min break between tasks. Four children were excluded as they did not complete the task. The children were divided into three age groups: 17 aged 33–36 months, 18 aged 37–39 months, and 21 aged 40–42 months. All children had normal visual acuity and no color blindness or weakness. Following completion of the experiment, each child received a reward. Informed consent was obtained from all parents.

### 2.2. Materials and Design

As shown in Figure 1, the materials were three-dimensional geometric boxes with two perceptual dimensions: shape (square, cylinder, and rectangle) and pattern (stripes, bricks, and dots). The purpose of employing patterns and shapes as the two dimensions of stimuli was to avoid shape bias, which had been observed in previous research on young children’s categorization. Children between 2 and 6 years old tend to view objects with similar shapes as belonging to the same category when asked to select an object of the same category as the target object [33,34]. We found no preference differences between children’s generalization of shapes and patterns in a simple categorization task [35]. Three colors were used to bind the patterns to avoid difficulty in discriminating them (e.g., yellow strips, green bricks, and red dots). In each set of tests, the experimenter first provided the child with a pair of boxes, indicated that one of the boxes contained a candy, and then asked the child to select the target box containing the hidden candy. After recording the children’s responses, the experimenter presented a new pair of boxes to the children to identify the hidden candy. Positive or negative feedback was provided at each step of the response. Each child was encouraged to form an initial rule, switch rules, and generalize the rules learned based on the feedback.

### 2.3. Procedure

#### 2.3.1. Practice Session

Children were tested individually in a quiet room. There were two practice trials in which the children were presented with two identical boxes: one containing candy and the other without. The experimenter explained the task requirements: “Hello, we will now play a game of searching for candies. Whenever we show two boxes, only one contains candy. Please use your finger to point out the box that you believe contains candy. However, do not touch the box.” Following the children’s choices, the researcher provided feedback by shaking the boxes. If the sound occurred indicating that candy was in that box, the children were praised: “Excellent! You got it.” If there was no sound, indicating no candy in that box, the children were encouraged: “Okay, let’s try again.”

#### 2.3.2. Test Session

The test was divided into three phases comprising nine steps (Figure 2 and Figure 3). The first phase was dedicated to establishing an initial rule that allowed the children to make rule-based choices. In Step 1 (free selection), the experimenter presented the child with a pair of boxes that differed in pattern and shape dimensions and instructed the child to select the box containing candy.

In Step 2 (rule formation), the initial rule was established, which adhered to the shared attributes of the two selected boxes (e.g., a child could assume that all cylindrical boxes contained candy). No feedback was provided during the first two steps, regardless of whether the child’s choice was correct. In Step 3 (rule reinforcement), the formed rule was confirmed by the choice between two new boxes, in which one dimension of the box matched the initial rule (e.g., had a shared shape with the boxes selected in the first step), whereas the other did not. After completing Step 3, the child was allowed to enter the second phase. If the child made an incorrect choice, they returned to Step 1 to reformulate the rules (see Figure 3).

The second phase was the rule-switching phase, which included Steps 4, 5, and 6. In Step 4 (negative feedback), the experimenter displayed the box the child had chosen in Step 3, along with a new box that varied in other dimensions. Children were likely to select a box identical to that selected in Steps 2 and 3. When a child chose a box that was identical to the boxes selected in the preceding two steps, the experimenter gave negative feedback and said, “Oh no, the candy has magically hidden in another box, so take a closer look at what kind of box it is.” A few seconds later, the experimenter asked the child to close their eyes. In each step before and after this, the children would assume that the box pair was new because the boxes were brought from under the table to the tabletop for display, even though some of the box features remained the same. However, in this step, the researcher asked the children to close their eyes and the researcher exchanged the spatial positions of their boxes. This arrangement enabled the children to understand that the boxes were not new, and thus they were more receptive to feedback that the candy was now in box with another feature, thus completing rule switching. Next, the researcher told the children to open their eyes and pick out the box containing the candy. If the choice was correct, then Step 5 (rule switching) proceeded; otherwise, Step 4 was repeated. The box pairs in Step 4 were again used in Step 5 to determine whether the children shifted their attention entirely to the new rule. If the children did this correctly, they proceeded to Step 6 (new rule reinforcement); otherwise, Step 4 was repeated. In Step 6, one of the boxes was identical to the target box in Step 5, whereas the other was a distracting box with a different attribute (e.g., a box with a new pattern).

In the third phase, children were encouraged to apply the newly learned rule (e.g., the pattern rule) to boxes that differed in other dimensions. In the example depicted in Figure 2, the boxes presented in Step 7 (rule generalization 1) are two cubes that differ in shape from the boxes in the preceding three steps; however, one box shares the same pattern (brick pattern) as the target boxes. It is necessary to note that in the second phase, the new rule in the minds of the children may be either a composite rule (e.g., a box with a cylindrical shape and a dot pattern contains candy) or a single rule (e.g., a box with a dot pattern contains candy). If it were the composite rule, it would be necessary to implicitly inhibit the interference from the shape (i.e., cylindrical shape) associated with the old rule when children generalize the new rule to boxes that vary in shape in Step 7. In contrast, if there was a single rule, there was no need for inhibition in Step 7. If the children correctly selected the target box, they proceeded to the final two steps.

### 2.4. Scoring and Statistical Analysis

To make the results more reliable, each child was tested on two sets of tasks with different materials, with a break of 5–10 min. There are two types of scores for each task set. One was the number of trials and errors per step, that is, the number of errors before a child made the correct choice, with a minimum score of 0 points and an uncapped maximum score. The greater the number of trials and errors, the lower the ability to learn from the task experience [36,37]. The second was the task score, which quantified the performance in each step (excluding the initial two steps) by awarding a score if the child correctly selected the target box without making any errors, with a minimum score of 0 and a maximum score of 2. In the DCCS task, a child was scored as having passed the post-switch phase if they correctly sorted at least five of the six cards [19]. With reference to the DCCS, we regarded the latter five steps following Step 5 (rule switching) as the post-switching phase, and a child was scored as having passed one set of the sRIT if they responded correctly in at least four of those five post-switch steps (i.e., Steps 5–9).

We used SPSS 23.0 statistical software for the statistical tests. To determine if a child passed the sRIT solely by guessing, we computed the proportion of children who completed Steps 5 and 6 correctly. Subsequently, an assessment of the overall pass rate was conducted to explore the appropriateness of the task as a measure of CF for children aged about three years. ANOVA and the Kruskal–Wallis test were used to explore potential within-subject differences between the ages to explore the developmental patterns observed with children’s CF. Finally, a partial correlation analysis was conducted to examine the relationship between CF and implicit inhibition.

## 3. Results

There was a positive correlation between the two sets of tasks (r = 0.268, *p* = 0.046); hence, the scores for the two sets of tasks were merged. After learning a rule in the first phase, in the second phase, 87.5% of the children scored more than one point in Step 5 (rule switching), and those who chose correctly in Step 5 also had a high probability of being correct in Step 6 (57 out of 78 sets were correct, 73.08%), which was significantly higher than guessing (*p* < 0.001).

The pass rate of the sRIT for each age group is shown in Figure 4. There were marginally significant differences in the pass rates across the three groups when a pass was defined as obtaining one score, that is, passing one or two sets of the sRIT (Fisher’s exact *p* = 0.054). Among the children who achieved a score in passing the sRIT, the number who passed the second set of tasks was higher than those who passed the first set of tasks, implying that there may be a practice effect (*χ*^2^ (df = 1 n = 23) = 9.783, *p* = 0.002). Therefore, we defined a pass as a child receiving two scores (i.e., passing both sets of tasks) and a failure score of zero. Approximately 20% of children under the age of three successfully completed both sets of the sRIT, whereas approximately 50% of children older than three years passed both sets of tasks. There was a significant difference in the pass rates between the two groups (Fisher’s exact *p* = 0.035).

The average number of trial-and-error operations for rule switching (Step 5), rule generation (Step 7), and all post-switch steps are listed in Table 1. One-way ANOVA showed a significant effect of age in Step 5 (F = 9.200, *p* < 0.001, η^2^ *p* = 0.258), Step 7 (F = 3.591, *p* = 0.034, η^2^ *p* = 0.119), and all post-switch steps (F = 5.118, *p* = 0.009, η^2^ *p* = 0.162). Post hoc tests showed that the number of trials and errors was significantly higher in the 33- to 36-month group than in the 40- to 42-month and 37- to 39-month groups (all *p* < 0.05).

The average scores for Steps 5 and 7 and all post-switch steps are shown in Table 2. The Kruskal–Wallis test revealed an age effect on the score at Step 5 (χ^2^ (df = 2, n = 56) = 7.596, *p* = 0.022), showing that the performance of rule switching increased as age increased. Further tests showed that the scores of the 40- to 42-month group (M = 1.67, SD = 0.483) were significantly higher than those of the 33- to 36-month group (M = 1.00, SD = 0.791, *p* < 0.05). When we used age three as a grouping criterion, the Mann–Whitney U tests on scores showed that children older than three outperformed children younger than three, as evidenced by the group difference at Steps 5 (Mann–Whitney U = 199.5, *p* = 0.009) and 7 (Mann–Whitney U = 228, *p* = 0.025).

Finally, the relationships between rule switching (Step 5) and rule generalization (Step 7) involving the implicit inhibition of the previous rule were assessed using partial correlation coefficients, and age was treated as a control variable. The results indicated that these two steps were positively correlated with score (r = 0.305, *p* = 0.023) and number of errors (r = 0.393, *p* = 0.003).

## 4. Discussion

Previous studies mainly adopted the DCCS task to measure CF [9,10,11]. Children must switch their attention to the new rule while inhibiting the old rule, which requires working memory for goal maintenance of the post-switch rule and inhibition of the pre-switch rule. It is challenging for 3-year-old children to simultaneously complete tasks containing two or more executive-function components; hence, most 3-year-old children cannot pass the DCCS task [38,39].

Based on studies that used the DCCS task to investigate the CF of children aged 3–4 years, we devised the sRIT, which decreases cognitive demands in a stepwise manner, making it more suitable for children around three years old, a possible time point of fast development in CF (e.g., [30,40]). The sRIT is comparable to an adapted version of the DCCS task [41], in which the irrelevant dimensions of the cards are fixed. In the sRIT, we fixed the dimensions of the old rule (e.g., presenting stimuli in the form of cylindrical boxes with bricks and dot patterns and asking the children to pay attention to a particular pattern or color). The main difference in the sRIT is that the rule-switching process is separate from the inhibition process. In the sRIT, we did not ask children to inhibit the old rule when they switched rules; instead, children were instructed to inhibit the old rule implicitly after they had successfully shifted their attention to the new rule. Specifically, in the second phase of the sRIT, a child was instructed to focus on the attribute (e.g., the brick pattern) associated with the new rule, while the previous rule (e.g., shape rule) remained intact. At this point, the new rule complements the old rule, and the two rules are compatible, ensuring that there is no need to inhibit the old rule during rule switching.

First, the reliability of the sRIT was demonstrated by a significant correlation between the two sets of tests. However, the lower correlation coefficients may be attributed to two potential factors: the shorter attention duration of the children and the greater individual differences. The statistical results confirmed that the children’s poor performance was a consequence of their inability to concentrate on the second set of questions because of the extended time required to complete the two sets of tasks consecutively. Another potential explanation is that the number of participants was insufficient to account for the significant individual differences among children in the younger age group. Consequently, in order to confirm the findings of this study, future investigations should either expand the sample size or administer tests to older children.

While the scoring criteria for the sRIT align with those commonly employed in categorization tasks, such as the DCCS, the scoring process in the sRIT is comparatively more challenging. Obtaining a perfect score in each step of the repeated trial-and-error process indicates a child’s complete comprehension of the rules and successful performance in rule switching. Additionally, having no persistent errors throughout the task indicates that the child has high CF [11,12]. However, we found that the majority of children in the three-year-old group passed more than one set of the sRIT, and about 50% of the children passed both sets of the sRIT, surpassing the pass rate in the standard version of the DCCS task for the same age group (36.33%) [20]. Consistent with the findings of previous studies [30,42], this result suggests that a task with less demand for inhibition is appropriate for assessing the rule-switching abilities of three-year-olds who are yet to fully develop inhibitory control [43].

Furthermore, our findings illustrate the multifaceted nature of CF. Two processes require the inhibition of components when children switch between the two rules on a range of preschool CF measures: resolution of response conflict and interference with the pre-switching rule. Most studies have concluded that the period between the ages of three and four is critical for the development of CF in children [9,24,30]. This is likely because children at this age possess well-developed inhibitory abilities that allow them to complete the switching task. Children aged 2.5–3 years also perform well in the distraction SwIFT when the task eliminates response conflict, thereby reducing the need for inhibition. The sRIT induces children to switch their attention to the new rule while maintaining the pre-switching rule; that is, the pattern rule is introduced while retaining the shape dimensions. At this point, because both boxes are characterized by the pre-switching rule, there is no response conflict or interference from the pre-switching rule. The only required inhibition is the distraction caused by the non-target box, which is easily overcome by infants between the ages of 2.5 and 3.5. Consequently, this investigation demonstrates that CF develops significantly around the age of 3, highlighting the importance of studying the development of CF in younger children.

Infants in their first year of life display fundamental forms of executive function [44]; however, the core components of working memory, inhibitory control, and CF develop rapidly during preschool age [45,46]. Most previous studies using the DCCS have considered the age range of 3–5 years as one of the most critical periods for the development of CF [9,10,13,47]. Consistent with the findings of previous studies, children’s performance in rule switching (i.e., the score in Step 5) increased with age, even though we used a smaller age band. Nevertheless, our study found that children older than three years demonstrated higher performance than those below the age of three years, which implies that age three is a milestone in the development of CF when the requirement for inhibition is excluded.

Our results are consistent with the conclusion of Blakey et al. (2016) [30], which suggests that children’s ability to overcome distraction from previous rules when switching develops between 2 and 3 years. Cognitive development around age three has been linked to significant growth spurts in the prefrontal cortex (PFC) [48,49]. Although previous studies of brain mechanisms associated with DCCS suggest that the PFC region develops rapidly at 3–4 years of age, a similar function may be present in young children, but is supported by different brain regions, leading to an underestimation of its maturity [50,51,52]. Moriguchi and Hiraki used fNIRS to investigate the neural correlates of the DCCS task in young children. They found that for 5-year-old children, the DCCS task activated the lateral prefrontal areas, and for 3-year-old children who performed the DCCS task correctly, significant activation was observed in the right lateral prefrontal regions [53,54]. An fMRI study revealed that the greatest changes in functional connectivity occurred throughout the developmental period from 2 to 4 years of age [55]. Generally, brain structure and functional networks are already in place by the third year of life, undergoing reorganization and fine-tuning beyond three years of age (e.g., myelinization and white matter development) [56,57,58,59,60]. In brief, the PFC regions related to CF develop from two to three years of age, while both behavioral performance and PFC activation increase significantly after three years old [61].

We found a positive correlation between performance in rule switching and rule generalization: specifically, children with stronger inhibitory control were better able to switch on the sRIT. This result supports the view that inhibitory control helps children ignore or suppress task-irrelevant information when required to switch or update task sets, particularly younger children [30]. In Step 7 (rule generalization) of the sRIT, children would demonstrate the ability to implicitly inhibit their attention to the rule-irrelevant feature (e.g., shape) of stimuli and generalize the new rule (e.g., pattern/color) to other stimuli with different shapes; therefore, the ability of inhibitory control is required in this step. Nevertheless, we did not find a significant effect of age on score in Step 7, which could potentially be attributed to the small age span or high individual differences within the age bands.

Compared to children under three years of age, older children made fewer trial-and-error errors, which is distinct from PE in the DCCS task [11]. Trial and error at each step of the sRIT is a necessary type of self-regulation associated with CF [62] and can also serve as an indicator of the learning process, resembling a reversal-learning task, which involves the ability to alter behavior when reinforcement contingencies change and is believed to play an important role in behavior adaptation [37,63,64]. The decrease in the number of errors made by the children as their age increased indicates that their ability to monitor and correct errors through feedback improves [65,66].

The development of CF around the age of three implies that interventions for cognitive function early in life are of greater importance and can have educational implications. Noteworthy is that a body of research has posited that the optimum age for preschool enrollment is around age three because of the rapid development in brain function and the greatest gains in terms of the improvement in linguistic and cognitive abilities [67,68,69]. Interestingly, our results may support the view that advancing to kindergarten at the age of 2.5 may yield more substantial benefits for the development of executive function [70,71,72].

Although this study adopted a novel rule-learning task to evaluate the CF of three-year-old children and obtained new findings, there are some limitations. First, future studies should simplify the steps of this task to minimize attention distraction and improve task performance, especially for younger children who may struggle to maintain their attention on tasks. Moreover, the limited sample size of children assessed and the narrow age range failed to provide a comprehensive understanding of the developmental trajectory of children’s CF. Therefore, future studies should increase the sample size to include a broader range (e.g., 2.5–6) to reveal the development of CF in the sRIT. Finally, this study did not assess the inhibitory abilities of the children individually, resulting in a lack of statistical control. This should be addressed in future studies to more clearly reveal the development of rule switching when inhibition is controlled.

## 5. Conclusions

In conclusion, a stepwise rule-induction task was devised to reveal the development of CF around the age of three years. In this novel task, the cognitive processes of rule inhibition and rule switching are separated, and children are instructed to pay attention to the new rule while not paying attention to the old rule. The results showed that rule-switching performance increased with age, with a significant difference between children younger and older than age three, suggesting that age three might be a milestone in the development of CF, particularly in the ability to switch rules.

## Figures and Tables

**Figure 1 behavsci-14-00578-f001:**
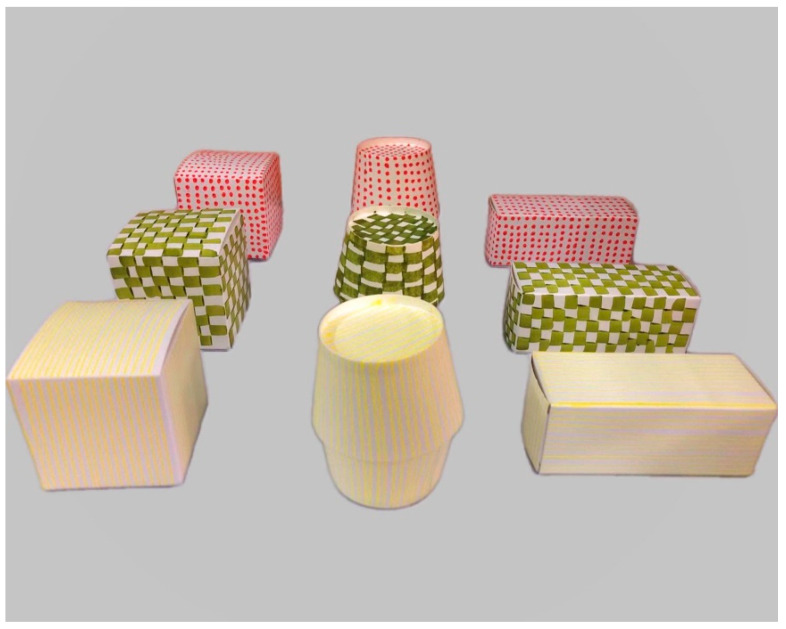
The sample stimuli used in the sRIT.

**Figure 2 behavsci-14-00578-f002:**
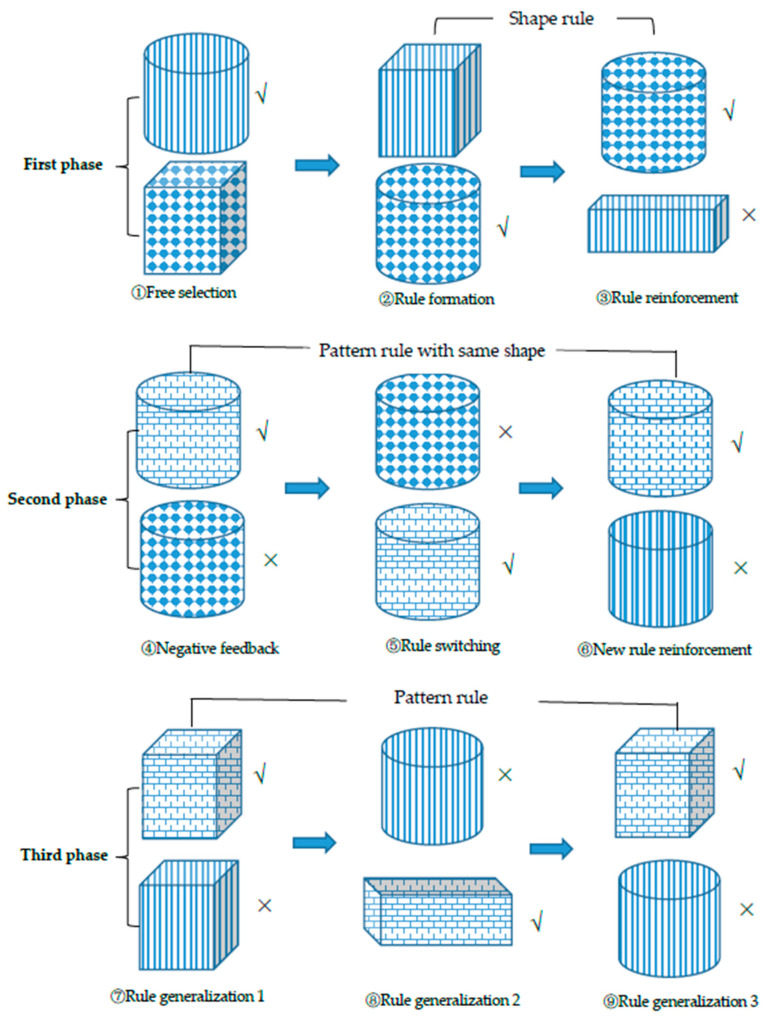
Sampled stimuli and the test procedure in a set of the sRIT. In the first phase, a child may select the cylindrical boxes and form the rule “the cylindrical boxes contain candy.” In Step 4, the most probable choice is identical to that selected in the preceding two steps. However, the experimenter provides negative feedback and indicates “the candy now is in another box.” In Steps 5 and 6, participants are expected to switch their attention to the new rule (i.e., the boxes with a brick pattern contain a candy). From Steps 7 to 9, children are expected to apply the new rule to boxes with different shapes. Checkmark “√” indicates positive feedback, while “×” indicates negative feedback.

**Figure 3 behavsci-14-00578-f003:**
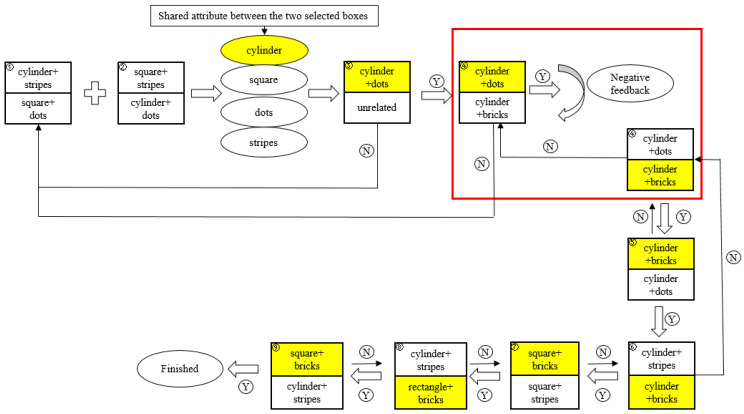
Flowchart of an example in the sRIT. Every step is represented by a pair of rectangular boxes containing exact attributes in the dimensions of pattern and shape. The color yellow indicates the correct choice. Y indicates that the child has made a correct choice, while N indicates that the child has made an incorrect choice. The arrow points to the next step.

**Figure 4 behavsci-14-00578-f004:**
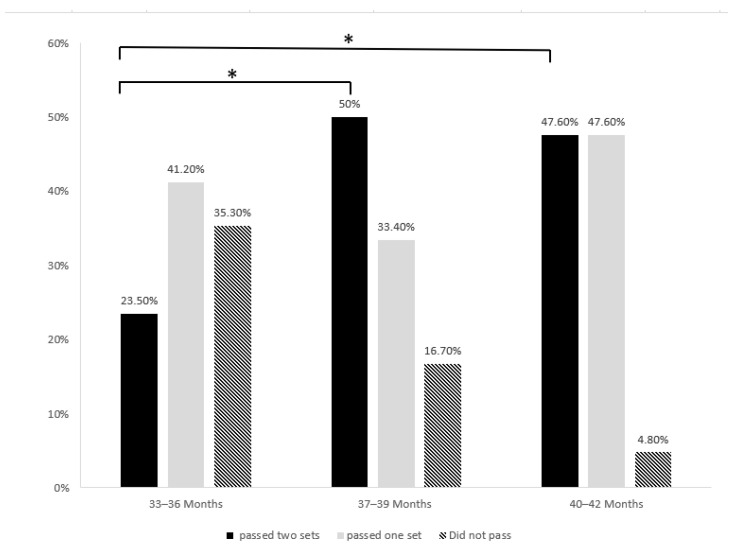
Percentage of children who passed the sRIT. * *p* < 0.05.

**Table 1 behavsci-14-00578-t001:** Number of trials and errors of Steps 4, 5, 7, and the post-switch (Steps 5–9) in each group.

	Age (Month)	M	SD	F	*p*
Step 5 (Rule switching)	33–36	2.29	2.392	9.200	<0.001 **
	37–39	0.67	0.840		
	40–42	0.38	0.590		
Step 7 (Rule generalization)	33–36	0.94	1.144	3.591	0.034 *
	37–39	0.33	0.594		
	40–42	0.29	0.644		
The total number of trial-and-errors of Steps 5–9	33–36	6.06	5.202	5.118	0.009 *
	37–39	3.22	2.315		
	40–42	2.86	1.526		

Note: * *p* < 0.05, ** *p* < 0.01.

**Table 2 behavsci-14-00578-t002:** Average scores of Steps 4, 5, 7 and the total score of post-switch (Steps 5–9) in each group.

	Age (Month)	M	SD	χ^2^	*p*
Step 5 (Rule switching)	33–36	1.00	0.791	7.596	0.022 *
	37–39	1.44	0.705		
	40–42	1.67	0.483		
Step 7 (Rule generalization)	33–36	1.29	0.772	5.328	0.070
	37–39	1.67	0.594		
	40–42	1.76	0.539		
The total score of Steps 5–9	33–36	6.47	2.125	3.421	0.181
	37–39	7.50	1.917		
	40–42	7.67	0.913		

Note: * *p* < 0.05.

## Data Availability

The raw data supporting the conclusions of this article will be made available by the authors upon request without undue reservation.

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
