# Peer review of "Age Three: Milestone in the Development of Cognitive Flexibility"

_behavsci, 2024, doi:10.3390/bs14070578_

Round 1

Reviewer 1 Report

Comments and Suggestions for Authors

This paper presents some positive and interesting results identifying age 3 as an important milestone in the development of cognitive flexibility, utilizing a developmentally appropriate rule induction task. Generally, I find the paper to merit publications though there are some places where the background and method could be clearer. More specific comments follow.

p. 2, line 52-55, I understood the rule (eventually), but this could/should be reworded (perhaps separating the rules of each switch condition and the required responses into two sentences).

p. 2, line 57, define A-not-B paradigm in brief

p. 2, lines 80-92, the level of detail here is better reserved for the method 

p. 4, line 148, not clear what "otherwise went back" indicated (in the event of a failure, they went back to step one?)

p. 6, line 195-6, what were the minimum and maximum possible scores for task score?

p. 6, line 200, is there a reason you adopted an 80% passing criterion (consistent with prior practice, for some theoretical reason), and have you examined what might happen with a more lenient criterion (e.g., 3 of 5 correct)?

p. 10, line 327, while its possible that the limited age span is a reason that you didn't find an effect of age, another reason could be high individual differences within your age bands (although I suppose one could eventually tie that back to limited age span as well, if those high individual differences are the result of small differences in age producing high variability in performance, due to the rapidly developing nature of cognitive ability at this age). 

Comments on the Quality of English Language

throughout introduction and method: tense shifting frequently. use past tense to refer to the method and findings of this or any study in particular, reserving present tense for statements of truth.

Author Response

Comments 1: p. 2, line 52-55, I understood the rule (eventually), but this could/should be reworded (perhaps separating the rules of each switch condition and the required responses into two sentences).

Response: We are grateful for your comments on our manuscript. Based on your suggestion, we reworked this statement as follows:

“In the “control” condition, children were told that the names of the shape stimuli attending school were their colors and were asked to name the colors (blue, yellow, red) of the stimulus. In the “inhibit” condition, some stimulus figures were shown with happy faces, whereas the remaining had a sad or frustrated expression. Children were told to name the colors of the happy-faced stimuli, while inhibiting those associated with the unhappy-faced stimuli. In the “switch” condition, some stimulus figures were wearing hats. The child was told that, now in this classroom, the names of the figures wearing hats were their shape (e.g., square or circle), and that the names of the figures without hats retained their color (e.g., blue, yellow, or red). The Shape School task is deemed unsuitable for children aged three years because of the comparatively high requirements of inhibition in the “switch” condition.” (see page 2 line 49-59)

Comments 2: p. 2, line 57, define A-not-B paradigm in brief

Response: Thank you for pointing this out. We agree with this comment. We added a little detail, and the revised sections are shown below.

“The A-not-B task, a classical paradigm used to investigate the development of many critical early cognitive abilities, involves hiding a desirable object at location A for several trials and then hiding it at a new location B. Several studies have used this task to demonstrate the emergence of CF in infants during their first year of life” (see page 2 line 60-63)

Comments 3: p. 2, lines 80-92, the level of detail here is better reserved for the method

Response: We appreciate the reviewers' suggestive comments very much. In order to give the paper the right amount of detail, we have cut this part of the methodology presentation and placed it among the methods. The revised sections are shown below.

“The issue to be addressed in our study is how CF develops around 3-year-olds if rule inhibition is not excluded from the task but merely separated from rule switching in the temporal dimension. To achieve this, a stepwise rule-induction task (sRIT) was devised and modified from the DCCS task. In sRIT, the post-switching phase is divided into two sub-processes: rule switching and rule inhibition. After learning a rule, children were first instructed to switch to the new rule while the pre-switching rule (i.e., the old rule) remained, and then the old rule was inhibited implicitly. As shown in Figure 2, sRIT comprises three phases with nine steps. In the first phase, children select the target based on their preferred rules. For instance, they may prefer a shape rule and assume that boxes with cylindrical shapes contain targets. In the second phase, they shift their attention to the new rule (e.g., pattern/color) through negative feedback while the previous rule (e.g., shape) remains intact; therefore, inhibiting the old rule is not necessary for rule switching. The third phase involves inhibiting or ignoring the old rule and generalizing the new rule to other stimuli of different shapes.” (see page 2-3 line 84-97)

Comments 4: p. 4, line 148, not clear what "otherwise went back" indicated (in the event of a failure, they went back to step one?)

Response: We revised it as below.

“if the child made an incorrect choice, they returned to Step 1 to reformulate   the rules (see Figure 3).” (see page 4 line 158-159)

Comments 5: p. 6, line 195-6, what were the minimum and maximum possible scores for task score?

Response: We added the details as below.

with a minimum score of 0 and a maximum score of 2.” (see page 6 line 257-258)

Comments 6: p. 6, line 200, is there a reason you adopted an 80% passing criterion (consistent with prior practice, for some theoretical reason), and have you examined what might happen with a more lenient criterion (e.g., 3 of 5 correct)?

Response: Our scoring criteria is consistent with the scoring criteria of the DCCS task.

In addition, no statistical analyses have been conducted with lenient criteria, as the current criteria referenced to DCCS have already have positive results. However, future studies could be conducted to investigate the potential outcomes.

Comments 7: p. 10, line 327, while it’s possible that the limited age span is a reason that you didn't find an effect of age, another reason could be high individual differences within your age bands (although I suppose one could eventually tie that back to limited age span as well, if those high individual differences are the result of small differences in age producing high variability in performance, due to the rapidly developing nature of cognitive ability at this age).

Response: We agree with this comment. Therefore, we have revised the discussion sections are shown below.

“Nevertheless, we did not find a significant effect of age on the score in Step 7, which could potentially be attributed to the small age span or high individual differences within the age bands.” (see page 11 line 410-412)

4. Response to Comments on the Quality of English Language

throughout introduction and method: tense shifting frequently. use past tense to refer to the method and findings of this or any study in particular, reserving present tense for statements of truth.

Response: Thank you very much for the suggestion to improve the quality of English language. In revision, the errors in language have been inspected and corrected by a Native English speaker.

Reviewer 2 Report

Comments and Suggestions for Authors

The topic addressed in this research is very relevant given that precisely one problem in the area is to have tests that allow for the assessment of cognitive flexibility. Moreover, assessing them at such early ages is also very necessary and there are few instruments that allow this. Therefore, the proposal is interesting and relevant for the area. However, some adjustments need to be made to the current version of the manuscript.

Introduction

1.- On page 2, line 71 it is mentioned "A few studies demonstrated that 2- to 3-year-old children can switch from old to new rules without demanding rule inhibition", I consider it important to make explicit the issue of the commitment of inhibitory control in the execution of the task. This is because this component will be relevant for the discussion of the manuscript and also because it is a variable that should have been statistically controlled.

2.- On the same page 2, but on line 73 it is mentioned: "The issue to be addressed in our study is how cognitive flexibility develops around 3-year-olds if rule inhibition is not excluded from the task but merely separated from rule switching in the temporal dimension". Since this is one of the strong ideas of the proposal, it should be explained in more depth and detail. Especially because it is the value of this novel evaluation approach.

3.- Page 2, line 81. It mentions: "they choose another box with the identical dimensions as the box selected in the first step". It would be important that lines before that it is made explicit which are the dimensions of possible categorisation. As far as I understand those are shape, pattern/colour. I don't understand why they didn't leave only colour as categories. I consider that pattern is a complex quality to discriminate and therefore the use of distinctive colours for each pattern would be a more parsimonious solution. In this sense, it could be that in the end the children could only discriminate two qualities (shape and pattern/control), which effectively decreases the cognitive load in the execution of the task and is appropriate for the age of the participants.

4.- Page 2, line 87 it is not clear to me in what form encourages is given: “encourages children to note that the target box is now hinged on the appearance of another dimensional feature that is rule-irrelevant in the preceding steps”.

5.- Page 2, line 90, On the fifth stage, explain how the new rule can be discovered or identified. Can the child himself choose the attribute, so if it is not shape it has to be pattern, is it? How do you get it to be the same figure and the same pattern?

6.- Page 2 line 94 the WCST is mentioned, since it is the first time it appears in the text, it should be explained that it is the Wisconsin card sorting test.

7.- Page 4, líne 154, Says “Oho” must say “Oh no, “

8.- I don't understand why children has to close their eyes in the second phase.

9.- page 6, line 191, it is mentioned that each child will be assessed in two sets of the task, explain what this means, ¿if they are assessed with exactly the same material?, i.e. ¿the same exercise is repeated or if other materials are presented?. Further on, line 211 mentions "There is a positive correlation between the two sets of tasks", consider for discussion the low correlation of performance found.

10.- Page 9, line 285 in a very powerful idea that gives strength to the novelty of the task. Therefore, I suggest explaining a little more in detail and depth.

11.- Finally, I consider that an important limitation of the study is not having had the statistical control of inhibitory control as it was evaluated with another test. However, this does not detract from the viability of the manuscript. Therefore, my suggestion is to make it explicit within the limitations of the study.

Author Response

Comments 1: On page 2, line 71 it is mentioned "A few studies demonstrated that 2- to 3-year-old children can switch from old to new rules without demanding rule inhibition", I consider it important to make explicit the issue of the commitment of inhibitory control in the execution of the task. This is because this component will be relevant for the discussion of the manuscript and also because it is a variable that should have been statistically controlled.

Response 1: Many thanks for your constructive suggestions to improve our manuscript. In revision, the role of the inhibitory component in performing the task has been emphasized in the revised manuscript as follows:

“…the post-switch stimuli did not have a response conflict, because the incorrect response option did not match the prompt image on the previously relevant dimension.” (see page 2 line 68-69)

“…when cues consistent with the target stimulus were provided before rule switching, there-by fulfilling the requirement of reducing the inhibition of the old rule” (see page 2 line 74-75)

However, the majority of experimental tasks, such as the DCCS task, Shape School, and Conflict SwIFT, are challenging for children under the age of three who have yet to develop inhibition. The failure of young children in these tasks may be because they are required to inhibit not only the interference of the pre-switching rule but also the dimensional conflict caused by the non-target stimulus. In contrast, 2–3-year-old children can switch from the old rule to the new rule for tasks without demanding rule inhibition (see page 2 line 77-83)

Comments 2: On the same page 2, but on line 73 it is mentioned: "The issue to be addressed in our study is how cognitive flexibility develops around 3-year-olds if rule inhibition is not excluded from the task but merely separated from rule switching in the temporal dimension". Since this is one of the strong ideas of the proposal, it should be explained in more depth and detail. Especially because it is the value of this novel evaluation approach.

Response: Thank you very much for your constructive suggestion to improve our manuscript. Based on your suggestion, we have revised this sections as below.

 “The issue to be addressed in our study is how CF develops around 3-year-olds if rule inhibition is not excluded from the task but merely separated from rule switching in the temporal dimension. To achieve this, a stepwise rule-induction task (sRIT) was devised and modified from the DCCS task. In sRIT, the post-switching phase is divided into two sub-processes: rule switching and rule inhibition. After learning a rule, children were first instructed to switch to the new rule while the pre-switching rule (i.e., the old rule) remained, and then the old rule was inhibited implicitly. As shown in Figure 2, sRIT comprises three phases with nine steps. In the first phase, children select the target based on their preferred rules. For instance, they may prefer a shape rule and assume that boxes with cylindrical shapes contain targets. In the second phase, they shift their attention to the new rule (e.g., pattern/color) through negative feedback while the previous rule (e.g., shape) remains intact; therefore, inhibiting the old rule is not necessary for rule switching. The third phase involves inhibiting or ignoring the old rule and generalizing the new rule to other stimuli of different shapes.” (see page 2 line 84-97)

Comments 3: Page 2, line 81. It mentions: "they choose another box with the identical dimensions as the box selected in the first step". It would be important that lines before that it is made explicit which are the dimensions of possible categorisation. As far as I understand those are shape, pattern/color. I don't understand why they didn't leave only color as categories. I consider that pattern is a complex quality to discriminate and therefore the use of distinctive colors for each pattern would be a more parsimonious solution. In this sense, it could be that in the end the children could only discriminate two qualities (shape and pattern/control), which effectively decreases the cognitive load in the execution of the task and is appropriate for the age of the participants.

Response: In previous studies on children’s categorization, preschoolers are susceptible to shape bias, which is a belief that objects with identical shapes are classified as the same category, that is, children always think the testing object shared shape with the target object is more likely to be the same category than the testing object shared color with the target object [33,34]. Nevertheless, We found that there was no significant difference between shape and pattern preferences in a simple categorization task [35]. In order to prevent the majority of children from inducing the rules by merely relying on shape bias in the rule learning phase of the sRIT, the pattern dimension is adopted and bound with color. For example, yellow is bound with stripes, green is bound with bricks, and red is bound with dots pattern. In the paper we have revised it accordingly, as follows.

“The purpose of employing patterns and shapes as the two dimensions of stimuli was to avoid shape bias, which had been observed in previous research on young children’s categorization. Children between 2–6 years old tend to view objects with similar shapes as belonging to the same category when asked to select an object of the same category as the target object [33,34]. We found no preference differences between children’s generalization of shapes and patterns in a simple categorization task [35].” (see page 3 line 120-125)

Comments 4: Page 2, line 87 it is not clear to me in what form encourages is given: “encourages children to note that the target box is now hinged on the appearance of another dimensional feature that is rule-irrelevant in the preceding steps”.

Response: The encouragement is provided through the negative feedback of the fourth step of the second phase, with the experimenter's expression, "The candy has magically hidden in another box." The child's attention might be drawn to the word "magic," and the experimenter would remind the child of the box's appearance. The child would then be asked to close their eyes, and the spatial positions of the boxes would be exchanged. Then, the child was instructed to open their eyes and select the box containing the candy.

““Oh no, the candy has magically hidden in another box, so take a closer look at what kind of box it is.” A few seconds later, the experimenter asked the child to close their eyes..” (see page 4 line 165-167)

Comments 5: Page 2, line 90, On the fifth stage, explain how the new rule can be discovered or identified. Can the child himself choose the attribute, so if it is not shape it has to be pattern, is it? How do you get it to be the same figure and the same pattern?

Response: Thank you for pointing this out. The target boxes in the step 5 and 6 were identical, with the difference being only in the non-target boxes. Therefore, we counted the percentage of correctness in two consecutive steps to determine whether the children truly understood the rules and successfully completed the switches. (see page 7 line 273-277)

Comments 6: Page 2 line 94 the WCST is mentioned, since it is the first time it appears in the text, it should be explained that it is the Wisconsin card sorting test.

Response: Thank you for pointing this out, and we have revised it as follows:

“Two predictions were made. First, the sRIT is a rule-learning task similar to the Wisconsin Card Sorting Test (WCST),” (see page 2 line 98-99)

Comments 7: Page 4, líne 154, Says “Oho” must say “Oh no, “

Response: We revised it as follows:

““Oh no, the candy has magically hidden in another box” (see page 4 line 165)

Comments 8: I don't understand why children has to close their eyes in the second phase.

Response 8: W have added more details are below.

“In each step before and after this, the children would assume that the box pair was new because the boxes were brought from under the table to the tabletop for display, even though some of the box features remained the same. However, in this step, the researcher asked the children to close their eyes and the researcher exchanged the spatial positions of their boxes. This arrangement enabled the children to understand that the boxes are not new and are more receptive to feedback that the candy is now in another feature-bound box, thus completing rule switching. Next, the researcher told the children to open their eyes and pick out the box containing the candy.” (see page 4 line 167-174)

Comments 9: page 6, line 191, it is mentioned that each child will be assessed in two sets of the task, explain what this means, if they are assessed with exactly the same material?, i.e. the same exercise is repeated or if other materials are presented?. Further on, line 211 mentions "There is a positive correlation between the two sets of tasks", consider for discussion the low correlation of performance found.

Response: We have written in more detail and added more discussion of the correlation results as below.

“To make the results more reliable, each child was tested on two sets of tasks with different materials, with a break of 5–10 minutes.” (see page 6 line 250-251)

First, the reliability of the sRIT was demonstrated by a significant correlation be-tween the two sets of tests. However, the lower correlation coefficients may be attributed to two potential reasons: the shorter attention duration of the children and the greater individual differences. The statistical results confirmed that the children's poor performance was a consequence of their inability to concentrate on the second set of questions because of the extended time required to complete the two sets of tasks consecutively. Another potential explanation is that the number of participants was insufficient to account for the significant individual differences among children in the younger age group. Consequently, in order to confirm the findings of this study, future investigations should either expand the sample size or administer tests to older children. (see page 9 line 336-345)

Comments 10: Page 9, line 285 in a very powerful idea that gives strength to the novelty of the task. Therefore, I suggest explaining a little more in detail and depth.

Response: Thank you for your suggestion. Accordingly, we have added more explanation as below.

“Consistent with the findings of previous studies [30,41], this result suggests that a task with less demand for inhibition is appropriate for assessing the rule-switching abilities of three-year-olds who are yet to fully develop inhibitory control [42].

Furthermore, our findings illustrate the multifaceted nature of CF. Two processes re-quire the inhibition of components when children switch between the two rules on a range of preschool CF measures: resolution of response conflict and interference with the pre-switching rule. Most studies have concluded that the period between the ages of three and four is critical for the development of CF in children [9,24,30]. This is likely because children at this age possess well-developed inhibitory abilities that allow them to complete the switching task. Children aged 2.5–3 years also perform well in the distraction SwIFT when the task eliminates response conflict, thereby reducing the need for inhibition. The sRIT task induces children to switch their attention to the new rule while maintaining the pre-switching rule; that is, the pattern rule is introduced while retaining the shape dimensions. At this point, because both boxes are characterized by the pre-switching rule, there is no response conflict or interference from the pre-switching rule. The only required inhibition is the distraction caused by the non-target box, which is easily overcome by infants between the ages of 2.5 and 3.5. Consequently, this investigation demonstrates that CF develops significantly around the age of 3, highlighting the importance of studying the development of CF in younger children.” (see page 9-10 line 354-373)

Comments 11: Finally, I consider that an important limitation of the study is not having had the statistical control of inhibitory control as it was evaluated with another test. However, this does not detract from the viability of the manuscript. Therefore, my suggestion is to make it explicit within the limitations of the study.

Response: We agree with this comment. We have revised it in revised manuscript as below.

Finally, this study did not assess the inhibitory abilities of the children individually, resulting in a lack of statistical control. This should be addressed in future studies to more clearly reveal the development of rule switching when inhibition is controlled.” (see page 11 line 445-448)
